# Development of a dual antigen lateral flow immunoassay for detecting *Yersinia pestis*

**Derrick Hau** **, Brian Wade, Chris Lovejoy, Sujata G. Pandit, Dana E. Reed, Haley L. DeMers, Heather R. Green, Emily E. Hannah, Megan E. McLarty, Cameron J. Creek, Chonnikarn Chokapirat, Jose Arias-Umana, Garett F. Cecchini, Teerapat Nualnoi, Marcellene A. Gates-Hollingsworth, Peter N. Thorkildson, Kathryn J. Pflughoeft, David P. AuCoin** \*

Department of Microbiology and Immunology, University of Nevada, Reno School of Medicine, Reno, Nevada, United States of America

\* daucoin@med.unr.edu

**Data Availability Statement:** All relevant data are within the manuscript and its Supporting Information files.

## Abstract

### Background

*Yersinia pestis* is the causative agent of plague, a zoonosis associated with small mammals. Plague is a severe disease, especially in the pneumonic and septicemic forms, where fatality rates approach 100% if left untreated. The bacterium is primarily transmitted via flea bite or through direct contact with an infected host. The 2017 plague outbreak in Madagascar resulted in more than 2,400 cases and was highlighted by an increased number of pneumonic infections. Standard diagnostics for plague include laboratory-based assays such as bacterial culture and serology, which are inadequate for administering immediate patient care for pneumonic and septicemic plague.

### Principal findings

The goal of this study was to develop a sensitive rapid plague prototype that can detect all virulent strains of *Y. pestis*. Monoclonal antibodies (mAbs) were produced against two *Y. pestis* antigens, low-calcium response V (LcrV) and capsular fraction-1 (F1), and prototype lateral flow immunoassays (LFI) and enzyme-linked immunosorbent assays (ELISA) were constructed. The LFIs developed for the detection of LcrV and F1 had limits of detection (LOD) of roughly 1–2 ng/mL in surrogate clinical samples (antigens spiked into normal human sera). The optimized antigen-capture ELISAs produced LODs of 74 pg/mL for LcrV and 61 pg/mL for F1 when these antigens were spiked into buffer. A dual antigen LFI prototype comprised of two test lines was evaluated for the detection of both antigens in *Y. pestis* lysates. The dual format was also evaluated for specificity using a small panel of clinical near-neighbors and other Tier 1 bacterial Select Agents.

### Conclusions

LcrV is expressed by all virulent *Y. pestis* strains, but homologs produced by other *Yersinia* species can confound assay specificity. F1 is specific to *Y. pestis* but is not expressed by all

**Funding:** Funding from the Naval Research Laboratory contract ND0173-16-C-2003 (DA) supported the generation of mAbs and ELISA development. Funds from a Defense Threat Reduction Agency (DTRA) contract HDTRA1-16-C-0026 (DA) expanded the library of mAbs and allowed for the development and optimization of LFIs. The funders had no role in study design, data collection and analysis, decision to publish, or preparation of the manuscript.

**Competing interests:** I have read the journal's policy and the authors of this manuscript have the following competing interests: The goal of this research funded by Naval Research Laboratory contract ND0173-16-C-2003 and the Defense Threat Reduction Agency (DTRA) contract HDTRA1-16-C-0026 was to develop a diagnostic for plague. Currently the project is funded to transition these prototypes into an FDA approved diagnostic for plague through the MCDC contract # MCDC OTA W15QKN-16-9-1002.

virulent strains. Utilizing highly reactive mAbs, a dual-antigen detection (multiplexed) LFI was developed to capitalize on the diagnostic strengths of each target.

## Author summary

Immunoassays were developed for the detection of two *Y. pestis* proteins, LcrV and F1, which have been characterized as potential biomarkers of plague. A total of twenty-two high affinity mAbs were isolated from BALB/c mice immunized with recombinant LcrV, F1 and F1-LcrV fusion protein via hybridoma technology. MAbs were characterized by Western blots, ELISA, and surface plasmon resonance. Antigen-capture LFIs and ELISAs were developed using the mAbs and optimized for analytical sensitivity. Prototype LFIs were evaluated to detect LcrV and F1 in surrogate clinical samples. A multiplexed LFI detecting both LcrV and F1 was assessed against a panel of *Y. pestis* isolates, clinically relevant near neighbors, and other bacterial Select Agents indicating high assay specificity. The prototype immunoassays will now need to be validated with multiple clinical matrices (i.e., whole blood), patient samples, and a larger specificity panel.

## Background

Plague is a febrile illness caused by *Yersinia pestis*, a Gram-negative, nonmotile coccobacillus. The bacterium was responsible for the Black Death, which devastated over a third of Europe's population between 1347–1353 [1]. The Centers for Disease Control and Prevention classifies *Y. pestis* as a Tier 1 Select Agent due to its infectious nature, high mortality rates, threat to public health, and potential as a biothreat. The spread of *Y. pestis* is facilitated by small mammals and insect vectors. The bacterium is transmitted to humans through flea bites, contact with animal excretions, or inhalation of aerosolized droplets. The different routes of infection lead to three forms of plague: bubonic, pneumonic, and septicemic. Bubonic plague is easily diagnosed by the inflammation of lymph nodes resulting in the formation of painful swellings called buboes. Bubonic plague is the least fatal of the three forms, with a 40–70% case fatality rate (CFR) when left untreated; however, bubonic plague may develop into more serious forms of the infection [2]. Pneumonic and septicemic plague present with nonspecific flu-like symptoms, leading to death in as few as three days post-exposure [3]. The CFR for pneumonic and septicemic infections approach 100% when left untreated [2]. Time is a critical factor for treating plague as effective therapeutics must be administered within 20 hours from the onset of symptoms to ensure the best chance for patient survival [4].

*Y. pestis* is a zoonotic bacterium found in all geographical regions besides Oceania, with Madagascar and the Democratic Republic of Congo being primary hot spots for annual plague outbreaks [5,6]. Madagascar had accounted for 74% of all cases reported to the World Health Organization between 2010 and 2015 [7]. During this period, 200–700 cases were reported annually, mainly in the form of bubonic plague which is rarely transmissible human-to-human [8,9]. During the 2017 plague season, Madagascar reported a total of 2,417 cases of plague with a CFR of 8.6% [10]. Identification of the initial cluster of infections allowed for a proper response to prevent a larger epidemic [11]. This outbreak not only marked an increase in overall cases, but more importantly, an increased percentage of pneumonic infections (70% of total infections) [8,11]. The increase in pneumonic infections may, in part, be attributed to human-to-human transmission that occurred via infectious droplets [12–14]. This along with

high mortality rates of pneumonic infections warrant the need for development of additional countermeasures for plague.

The gold standard for diagnosing plague is bacterial culture [15]. Patient serology can be used to indicate if an individual was infected with *Y. pestis;* however, this method is hindered by its capacity to detect active infections as an antibody response is delayed upon exposure and a positive result may be due to a previous exposure [15–17]. These methods can be time consuming and require specialized laboratories and trained technicians. Advancements in rapid diagnostic tests (RDT) such as LFIs have made the detection of *Y. pestis* more feasible in low-technology settings and are instrumental in controlling plague outbreaks in endemic regions. A comparative study on various diagnostic methods has deemed an LFI to be an ideal platform for diagnosing plague [18]. LFIs are rapid, membrane-based immunoassays using antibodies linked to gold-nanoparticles for visual detection. Temperature stable reagents and user-friendly protocols make LFIs ideal for resource limited settings. The current LFI used in Madagascar detects the F1 antigen and has a sensitivity range of 25–100% and a specificity range of 59–79% when compared to bacterial culture and PCR [19–23]. Of concern, however, is that F1⁻ strains, mutants that do not express the F1 pilus structure, have been identified in clinical as well as laboratory settings [19,24,25]. In the present study, we describe the initial evaluation of a prototype multiplexed RDT that allows for detection of two *Y. pestis* antigens.

*Y. pestis* is a facultative anaerobe equipped with several virulence factors to allow survival within macrophages and in extracellular spaces [26]. Genes encoding virulence factors are dispersed throughout the genome, and are also encoded on three plasmids (pCD1, pPCP1, pMT1) [27]. Current LFIs used to diagnose plague detect F1, a protein encoded by the *caf1* gene on pMT1, a plasmid unique to *Y. pestis* [18,25,28]. The 15.5 kDa F1 antigen polymerizes to form a filament capsule surrounding the bacterium, protecting it from phagocytosis [29]. Expression of F1 is temperature-induced at >33°C [28]. The antigen is known to be shed during infection, making it a viable candidate biomarker for diagnosing plague [30–32]. Though attenuated in its ability to cause bubonic plague, F1⁻ isolates remain highly virulent by the inhalation route leading to pneumonic infections [25,33].

The pCD1 plasmid carries genes encoding a type-III secretion system (T3SS) and its related effector proteins [34]. The pCD1 plasmid is found in clinically relevant neighbors *Y. pseudotuberculosis and Y. enterocolitica* [35]. The T3SS is associated with a low-calcium response crucial for pathogenicity of *Yersinia* species [34]. Low-calcium response V (LcrV) is a multifunctional protein serving as the needle tip of the T3SS to translocate *Yersinia* outer proteins into host cells [36]. LcrV is also translocated during the process and suppresses the host inflammatory response by upregulating interleukin 10 via Toll-like receptor 2 [37]. Mutations in *lcrV* lead to avirulence in mice due to the inability to translocate effector proteins [38,39]. Though LcrV is displayed on the surface of the bacterium and translocated via the T3SS, the antigen has also been reported to be shed into growth media *in vitro* [40]. A study conducted using a murine model of infection indicated that LcrV is detectable in serum and bronchoalveolar lavage fluid (BALF) in mice displaying symptoms of bubonic and pneumonic plague [41]. Since LcrV is essential for pathogenicity and appears to be shed during infection, it may serve as an important alternative biomarker for the diagnosis of plague [42–44].

While *Y. pestis* remains susceptible to many antibiotics, a growing number of resistant strains have been reported; and the ease of transferring resistance via plasmids is well established [45–47]. Furthermore, there is no FDA-approved vaccine for plague. A short incubation period, high mortality rates, and potential for mass infection through aerosolization warrant the development of novel countermeasures for the plague. In this study, prototype immunoassays were developed for the detection of LcrV and F1 for potential diagnosis of plague infections. Hybridoma cell lines producing mAbs against LcrV and F1 antigens were produced.

The resulting mAbs were evaluated for binding kinetics and used to develop sensitive immunoassay prototypes. Many mAb pair combinations were evaluated to develop LFIs and ELISAs for the detection of LcrV and F1. Surrogate clinical samples were used to determine the analytical sensitivity or LOD for the LFI prototypes. Pathogenic near neighbors and other Tier 1 bacterial Select Agents were used to begin to evaluate specificity of the LFI. The overall goal is for these prototypes to eventually be validated in a variety of sample matrices collected from plague patients followed by FDA approval.

## Methods

### Ethics statement

The use of Normal Human Serum has been reviewed by the Institutional Review Board at the University of Nevada, Reno (OHRP #IRB00000215). Normal Human Serum acquired from a commercial source has been classified as exempt human subject research (exemption 4) as i) no specimens will be collected specifically for this study and ii) there are no subject identifiers. As a consequence, the use of clinical samples in this project does not meet the criteria of human subject research as per 45 CFR 46 of the HHS regulations.

The use of laboratory animals in this study was approved by the University of Nevada, Reno Institutional Animal Care and Use Committee (protocol number 00024). All work with animals at the University of Nevada, Reno was performed in conjunction with the Office of Lab Animal Medicine, which adheres to the National Institutes of Health Office of Laboratory Animal Welfare (OLAW) policies and laws (assurance number A3500-01).

### Monoclonal antibody (mAb) production

Animals were used in this study to produce reagents (mAbs) for the development of immunoassay assays. All animal work and husbandry were in the animal facility at the University of Nevada, Reno. Twenty female CD1 mice, 6–8 weeks old (Charles River Laboratories, Inc.), were immunized with either recombinant F1/LcrV fusion protein (F1-V), LcrV, or F1 (Biodefense and Emerging Infections Research Resources Repository [BEI Resources], Manassas, VA). Subcutaneous injections of recombinant F1-V, LcrV, or F1 in emulsions of Titermax Gold Liquid Adjuvant (TiterMax, Norcross GA) were performed with subsequent boosts at weeks 4 and 8. Immunizations of recombinant F1 were also performed using Freund's complete adjuvant (MilliporeSigma, Billerica, MA) via intraperitoneal injection with subsequent boosts using Freund's incomplete adjuvant (MilliporeSigma) at weeks 4, 8 and 12. The number of animals was selected to account for variability in the immune response, human error, and for any unforeseeable causes (i.e. death of animal). Sera samples were collected via submandibular or retro-orbital survival bleeds and titers were screened by indirect ELISA. A final boost of recombinant protein without adjuvant was administered intravenously three days prior to splenectomy. Euthanasia was performed by $CO_2$ asphyxiation and final bleeds were performed via cardiac sticks. Splenocytes from all mice were harvested and hybridoma cell lines were produced by standard technique using the P3X63Ag8.653 fusion partner [48]. Splenocytes from mice with the highest titers against the target antigen, as assessed by indirect ELISA, were prioritized to produce hybridoma cell lines. Remaining splenocytes were frozen down and stored in liquid nitrogen. Hybridoma cell lines were isolated through cloning by limiting dilutions. Cells were grown *in vitro* using Dulbecco's Modified Eagle's Medium (DMEM) containing fetal bovine serum (FBS) and conditioned media. Supernatant was collected and purified by protein A affinity chromatography yielding >10 mg/L purified mAbs.

## Indirect ELISA

Microtiter plates (96-wells) were coated with recombinant protein (LcrV or F1) in phosphate buffered saline (PBS) overnight at room temperature. Plates were washed with PBS containing 0.05% Tween-20 (PBS-T) then blocked for 90 minutes at 37˚C with PBS-T containing 5% non-fat milk (blocking buffer). Primary antibodies (mouse sera or purified mAbs) were diluted in blocking buffer and serial two-fold dilutions were performed across plates. Primary antibodies were incubated for 90 minutes at room temperature. Plates were washed with blocking buffer then incubated with horseradish peroxidase (HRP) labeled goat anti-mouse IgG antibody (Southern Biotech, Birmingham, AL) for 60 minutes at room temperature. Isolated mAbs were also analyzed using IgG subclass specific (IgG3, IgG1, IgG2a, IgG2b) goat anti-mouse secondary antibodies (Southern Biotech) for further characterization. Plates were washed with PBS-T and incubated with tetramethylbenzidine (TMB) substrate (Kirkegaard & Perry Laboratories, Inc., Gaithersburg, MD) for 30 minutes at room temperature. An equal volume of 1 M phosphoric acid ($H_3PO_4$) was used to stop the reaction, and colorimetric data was read at an optical density of 450 nm ($OD_{450}$).

## Bacterial lysate preparation

At biosafety level 2 (BSL2), a glycerol stock of *Y. pestis* Harbin-35 (BEI Resources) was streaked onto a brain-heart infusion (BHI) agar plate and incubated at 28˚C for 48 hours. An individual colony was picked, inoculated into tryptic soy broth (TSB), and grown overnight at 37˚C shaking in 5% $CO_2$. Larger cultures were inoculated from the starter culture and grown for 48 hours at 37˚C shaking in 5% $CO_2$. Bacterial cells were pelleted by centrifugation and resuspended in PBS. Cells were heat-inactivated at 80˚C for 2 hours. Bacterial supernatant was 0.2 μm filtered. Bacterial cell lysate and supernatant were backplated onto BHI agar and incubated at 37˚C for at least 72 hours to ensure no viable cells were present.

Additionally, bacterial lysates for *Y. pestis* KIM D19, *Y. pestis* A12 Derivative 6, *Y. pseudotuberculosis* IP2666, *Y. enterocolitica* WA, *Francisella tularensis* B38, *F. tularensis* LVS, *Bacillus anthracis* Ames-35, *Burkholderia pseudomallei* K96243, *B. pseudomallei* 1026B, and *B. pseudomallei* Bp82 were prepared as per instructed (BEI Resources). Bacterial lysates were heat inactivated and separated by cells and supernatant. *B. pseudomallei* strains K96243 and 1026B were propagated at biosafety level 3 (BSL3) and confirmed nonviable before removal to (BSL2) by a validated protocol. *Y. pestis* KIM D19, *Y. pestis* A12 Derivative 6, *Y. pseudotuberculosis* IP2666, *Y. enterocolitica* WA, *F. tularensis* B38, *F. tularensis* LVS, *B. anthracis* Ames-35, and *B. pseudomallei* Bp82 lysates were propagated at BSL2 as per protocol above. Inactivated bacterial cells were resuspended in PBS to an optimal density at 600 nm ($OD_{600}$) of 0.5.

## Recombinant LcrV and F1 cloning and expression

Genes encoding LcrV and F1 were cloned into *Escherichia coli*, expressed and purified. The F1 encoding gene (*caf1) and lcrV* were amplified by polymerase chain reaction (PCR) from the *Y. pestis* Harbin-35 lysate, using primers shown in S1 Table. The *caf1* gene was void of the first 63 base pairs which encodes for a cleaved signal peptide [49]. Each gene was cloned into the pQE-30 Xa Vector (Qiagen, Hilden, Germany) by Gibson Assembly (New England Biolabs (NEB), Ipswich, MA). Plasmids were sequence verified and transformed into *E. coli* M15 for protein expression. *E. coli* containing expression plasmids were grown at 37˚C to an optimal density at $OD_{600}$ of 0.6 before inducing protein expression by the addition of isopropyl ß-D-1-thiogalactopyranoside (IPTG) at an end concentration of 1mM. Induced cultures were grown at 37˚C for 12–16 hours. Bacterial cell pellets were collected by centrifugation then lysed with

BugBuster 10X Protein Extraction Reagent (MilliporeSigma) and sonication. The soluble fraction for both were purified using Protino Ni-TED resin (Macherey-Nagel, Duren, Germany).

## Western blot

A standard Western blot procedure was performed using semidry blotting. 6x reducing or non-reducing Laemmli loading buffer was added to bacterial lysate. The reduced sample was then boiled for 10 minutes to denature the proteins. Samples were separated on a 10% sodium dodecyl sulfate (SDS) gel, and proteins were transferred to a nitrocellulose membrane (Bio-Rad Laboratories, Hercules, CA). HRP-labeled LcrV mAbs were used to probe the membrane directly. Unlabeled F1 mAbs were used to probe the membrane, followed by an HRP-labeled goat anti-mouse Ig for detection. Signal was developed using SuperSignal™ West Femto Maximum Sensitivity Substrate (ThermoFisher Scientific, Grand Island, NY) and imaged using a ChemiDoc XRS imaging system (Bio-Rad Laboratories).

## LFI

LFIs were initially constructed using Fusion-5 matrix membrane (GE Healthcare, Piscataway, NJ), FF120HP nitrocellulose membrane (GE Healthcare) and C083 cellulose fiber sample pad strips (Millipore Sigma). FF120HP membranes were striped with unlabeled antibodies using the BioDot XYZ platform (BioDot, Irvine, CA). Purified *Y. pestis* mAbs were striped at 1 mg/mL and served as the test line. Unlabeled goat anti-mouse Ig (Southern Biotech) were striped serving as the control line. All mAbs were conjugated to 40 nm colloidal gold particles (DCN Diagnostics, Carlsbad, CA), blocked with bovine serum albumin and concentrated to an optical density of 10 at 540 nm. Stability of the mAb conjugates were tested by stability in high salt conditions. Colloidal gold conjugated mAbs were spotted roughly 8mm from the top of the Fusion-5 matrix membrane prior to running the assay and served as the detection antibody. Initial screening was done using all possible mAb pairs for LcrV and F1. A single concentration of recombinant protein (100 ng/mL) was used to evaluate mAb pairing. Samples of 40 μL were loaded onto the sample pad then placed in a well containing 150 μL of chase buffer only. Each prototype was run in parallel with chase buffer only, as a negative control. Test line intensity was read promptly after 20 minutes using the ESE Lateral Flow Reader (Qiagen). Top performing pairs were then down selected using a concentration of 1 ng/mL of recombinant protein.

Top LFI prototypes were optimized for testing using human sera. Recombinant protein (LcrV or F1) was spiked into six lots of pooled normal human sera ([NHS], Bioreclamation IVT, Westbury NY & Innovative Research, Novi MI) and evaluated for signal intensity and non-specific background signal. LFI components optimized include conjugate release pads, nitrocellulose membranes, wicking pads, chase buffers, striping concentrations, gold conjugate diluents and sample pre-treatment steps (S2 Table). The final prototypes were constructed using UniSart CN140 nitrocellulose membrane (Sartorius, Germany), CFSP203000 absorbent pad, and conjugate pad grade 8951 (Ahlstrom, Finland). F127 and 10G surfactants were added to the gold diluent for the LcrV and F1 prototypes respectively to help with the release of the conjugate from the conjugate pad. Mouse IgG was added to the human serum samples as a pretreatment step to prevent human anti-mouse antibody (HAMA) interference [50].

## Antigen-capture ELISA

The top eight performing mAbs ranked by LFI testing were screened to develop antigen-capture ELISAs. Microtiter plates were coated overnight with 1 μg/mL capture mAb. After blocking, recombinant protein (F1 or LcrV) was diluted into blocking buffer and serial two-fold

dilutions were performed across plates. Detection antibody at 0.1 μg/mL was incubated for 60 minutes at room temperature. HRP labeling of mAbs was performed using EZ-Link Plus Activated Peroxidase (ThermoFisher Scientific). Plates were washed with PBS-T and incubated with TMB substrate for 30 minutes at room temperature. An equal volume of 1M $H_3PO_4$ was used to stop the reaction, and colorimetric data was read at $OD_{450}$.

Checkerboard ELISAs were performed using the top two mAb pairs to optimize concentrations of capture and detection mAbs. Optimization was performed using two-fold serial dilutions of either the capture or detection mAbs in independent experiments. The capture mAbs was first optimized by using concentrations ranging from 0.078–10 μg/mL with 1 μg/mL detection mAb. Capture mAb concentrations were chosen based on signal intensity and minimal background. The selected capture mAb concentrations were used to optimize the detection mAbs from a range of 0.0078–1 μg/mL. The optimized ELISA conditions were selected based on the lowest LOD defined as the concentration at two-fold background signal. The top two mAb pairs for each antigen were evaluated to determine the theoretical ELISA limit of detection defined as two-fold background signal.

## Surface plasmon resonance (SPR)

SPR experiments were conducted on the Biacore X100 instrument using the His Capture format (GE Healthcare). A CM5 chip surface was prepared using the His Capture Kit as per manufacturer's recommendation. For each cycle, his-tagged recombinant protein (LcrV or F1) diluted into HBS-EP+ buffer (GE Healthcare) was immobilized onto the anti-His capture surface. LcrV was immobilized at a concentration of 5 μg/mL and F1 was immobilized at a concentration of 1 μg/mL. Antigen capture was performed at 5 μL/second for 60 seconds followed by 120 second stabilization. These conditions were established for optimal kinetic analyses of each mAb. Full kinetic analyses were performed by injecting each purified mAb for 7 cycles at a concentrations range of 0.5–50 μg/mL over the LcrV or F1 surface for 120 seconds followed by a dissociation period of 240 seconds at 30 μL/second. The anti-His capture surface was regenerated between each cycle using 10 mM glycine pH 1.5 (GE Healthcare) for 30 seconds at 10 μL/second. Binding kinetics and affinity were evaluated using a bivalent model on the Biacore X100 Evaluation Software (GE Healthcare).

## Results

### Hybridoma production and mAb reactivity

In order to develop antigen-capture immunoassays for the detection of LcrV and F1, a large panel of mAbs were isolated and evaluated to determine the optimal conditions for assay performance. Titermax Gold or Freund's adjuvants were combined with antigens and used for immunization [51,52]. Twenty-two hybridoma cell lines that produce mAbs against *Y. pestis* LcrV or F1 were isolated (Table 1). MAb reactivity was confirmed by indirect ELISA to recombinant F1 or LcrV proteins. Additional mAbs were isolated from mice immunized with the F1-V fusion protein; however, these mAbs were only reactive to the F1-V fusion protein and not reactive to the individual proteins. Subclass specific secondary antibodies were used to characterize each mAb subclass, and all antibodies were determined to be members of the IgG1, IgG2b, or IgG2a subclass (Table 1). These antibody subclasses are preferred over the IgG3 subclass in the development of antigen-capture immunoassays as the IgG3 subclass can self-associate resulting in increased background signal and potential false positive reactions [53].

Western blot analysis probing *Y. pestis* Harbin-35 lysate determined mAb reactivity against the two *Y. pestis* proteins. High density bands were detected at the expected molecular weight

**Table 1. Monoclonal antibody (mAb) library against *Y. pestis* LcrV and F1 antigens.**

| Antigen | mAb | Immunization | Subclass |
| --- | --- | --- | --- |
| LcrV | 2B2 | F1-V | IgG2a |
| | 4E8 | LcrV | IgG2a |
| | 5D3 | LcrV | IgG1 |
| | 6E5 | LcrV | IgG1 |
| | 6F10 | LcrV | IgG1 |
| | 8F3 | LcrV | IgG1 |
| | 8F7 | LcrV | IgG2a |
| | 8F10 | LcrV | IgG1 |
| F1 | 3A2 | F1-V | IgG2a |
| | 3F2 | F1 | IgG1 |
| | 4E5 | F1 | IgG2a |
| | 4F12 | F1 | IgG1 |
| | 5E10 | F1 | IgG2a |
| | 9B7 | F1 | IgG1 |
| | 10D9 | F1 | IgG1 |
| | 10E3 | F1 | IgG1 |
| | 11B8 | F1 | IgG2a |
| | 11C7 | F1 | IgG1 |
| | 12B6 | F1 | IgG2a |
| | 12E10 | F1 | IgG2a |
| | 12F5 | F1 | IgG2b |
| | 15C4 | F1 | IgG2a |

of roughly 40 kDa indicating reactivity with monomeric LcrV protein (Fig 1A) [54]. Reactivity is observed at higher molecular weights, indicating that both monomeric and multimeric forms are produced in bacterial culture [54]. MAbs 4E8 and 5D3 had limited reactivity to the non-reduced protein as well as the multimeric forms suggesting reactivity to linear epitopes that are more available in reducing conditions (Fig 1B). MAbs isolated against F1 showed differing levels of reactivity to the monomeric F1 antigen in the reduced Western blot (Fig 2A). The non-reduced Western blot analysis showed reactivity of the F1 mAbs against the assembled F1 capsule which resulted in an expected laddering pattern (Fig 2B) [55]. Reducing conditions disrupt disulfide bonds as heat disrupts other protein-protein interactions; therefore these conditions disassemble the F1 capsule's native polymeric structure. MAb 3A2 was isolated from a mouse immunized with the F1-V fusion protein and showed preferential reactivity to the monomeric structure and had low reactivity to the assembled capsule. Inversely, mAbs 9B7 and 12F5 showed limited reactivity to the reduced bacterial lysate and were more reactive against higher molecular weight F1 multimers present in the non-reduced lysate (Fig 2B). Reactivity of mAbs 9B7 and 12F5 may suggest binding only occurs to a specific epitope formed in the F1 capsule between at least three or more subunits (>46.5 kDa).

## Binding affinity and kinetics by surface plasmon resonance (SPR)

Characterization of the mAbs in each library included analysis of binding kinetics by SPR. Experiments were performed in triplicate and evaluated using a bivalent binding model. The association rate ($k_a$), dissociation rate ($k_d$), and equilibrium dissociation constant ($K_D$; $K_D = k_d/k_a$) are reported in Table 2. LcrV mAbs displayed a narrow range of equilibrium dissociation constants (0.3–4.5 nM). F1 mAbs had a larger range of 0.002–250 nM. Interestingly,

## (A) Reduced

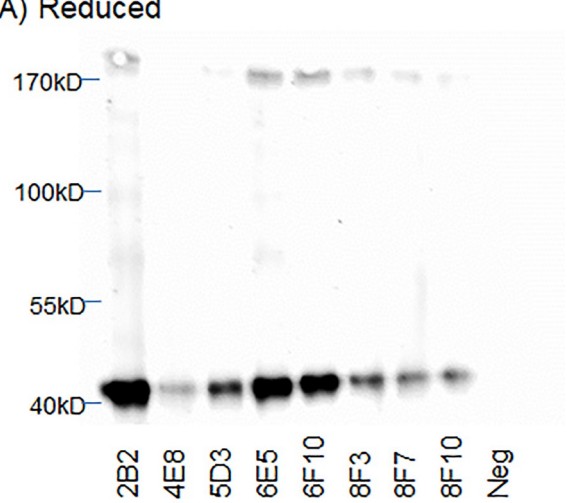

## (B) Non-reduced

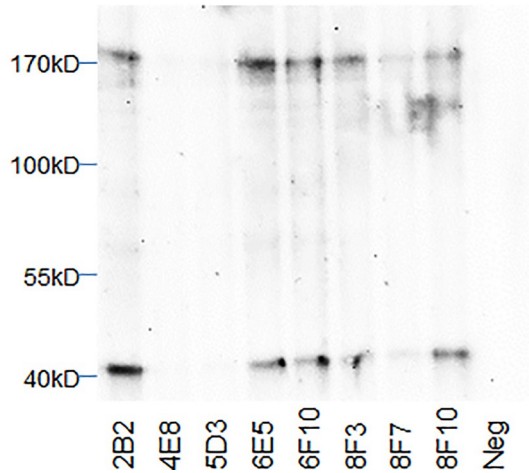

**Fig 1. Western blot analysis of anti-LcrV monoclonal antibodies (mAbs) against *Yersinia pestis* Harbin-35 lysate.**
Horseradish peroxidase (HRP) conjugated LcrV mAbs (1 μg/mL) were used to probe **(A)** reduced and **(B)** non-reduced *Y. pestis* Harbin-35 bacterial lysate (1.5x10^6 cells/lane) by direct Western blot.

mAbs 2B2 and 3A2 were isolated from mice immunized with the F1-V fusion protein. 2B2 displayed a high affinity to recombinant LcrV ($K_D$ = 2.6 nM); however, 3A2 displayed the poorest affinity to recombinant F1 ($K_D$ = 250 nM).

## LFI development and optimization

The library of mAbs were evaluated as capture and detection components for the development of LFI prototypes. MAb pairs were quantitatively ranked based on the signal intensity at the test line when analyzing a sample containing 100 ng/mL of recombinant protein (LcrV or F1) spiked into chase buffer minus nonspecific signal when chase buffer alone was analyzed (S3 and S4 Tables). Though LFIs are generally evaluated by visual detection, the use of Qiagen ESE Lateral Flow Reader provided a standardized measure to minimize human variability and

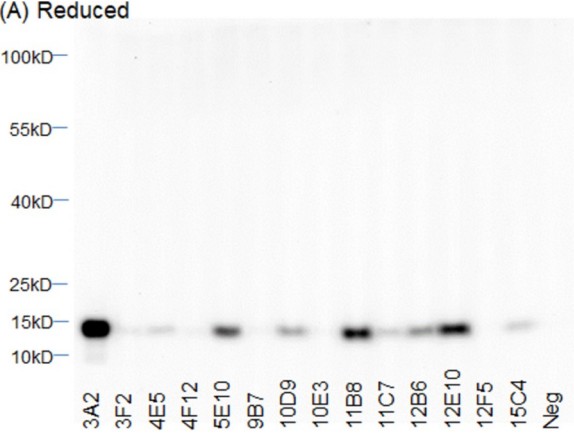

**Fig 2. Western blot analysis of anti-F1 monoclonal antibodies (mAbs) against *Yersinia pestis* Harbin-35 lysate.**
Anti-F1 mAbs (1 µg/mL) were used to probe **(A)** reduced and **(B)** non-reduced *Y. pestis* Harbin-35 bacterial lysate
(1.5x10$^6$ cells/lane) by indirect Western blot. HRP-conjugated goat anti-mouse Ig was used for detection of F1 mAb
binding.

error. Visual LOD is estimated to have an intensity between 15–30 mm$^*$mV among various
LFIs developed in our laboratory. The mAb pairs were then down selected for detection of 1
ng/mL recombinant protein (Table 3). The top four performing pairs (capture/detection) were
evaluated for detection of LcrV in bacterial lysate (S1 Fig). MAb pair 8F10/2B2 was highly
reactive to bacterial lysate. This pair, however, consistently had nonspecific reactivity with con-
trol, prompting the evaluation of the next top performing pair (8F10/6F10). Further optimiza-
tion is warranted to minimize nonspecific reactivity as these results together suggest 8F10/2B2
is more analytically sensitive when detecting native protein despite 8F10/6F10 having similar
results with recombinant protein (Table 3). The top pair for F1 detection was 11C7/3F2 when
evaluated with 1 ng/mL recombinant F1 protein (Table 3). S5 Table shows additional F1 LFI
prototypes that displayed reactivity in chase buffer alone.

Testing complex matrices such as human sera often lead to assay signal loss and nonspecific
binding described as matrix effects [56]. To account for interference of the matrix on down-
stream applications, the top LFI prototypes were further optimized for assaying human serum.
The LOD was defined as the minimum concentration of recombinant protein at which a

**Table 2. Affinity and kinetics analysis of _Y. pestis_ mAbs by surface plasmon resonance.**

| Antigen | mAb | $k_a$ x $10^3$ (M$^{-1}$s$^{-1}$) | $k_d$ x $10^{-3}$ (s$^{-1}$) | $K_D$ (nM) |
|---------|-----|-----------------------------------|------------------------------|------------|
| LcrV | 2B2 | 52 | 0.14 | 2.6 |
| | 4E8 | 210 | 0.78 | 3.7 |
| | 5D3 | 65 | 0.22 | 3.4 |
| | 6E5 | 120 | 0.54 | 4.5 |
| | 6F10 | 140 | 0.58 | 4.1 |
| | 8F3 | 93 | 0.17 | 1.9 |
| | 8F7 | 100 | 0.18 | 1.8 |
| | 8F10 | 250 | 0.074 | 0.3 |
| F1 | 3A2 | 5.6 | 1.4 | 250 |
| | 3F2 | 65 | 0.48 | 7.4 |
| | 4E5 | 250 | 0.39 | 1.6 |
| | 4F12 | 91 | 1.8 | 19 |
| | 5E10 | 230 | 0.12 | 0.5 |
| | 9B7 | 42 | 0.45 | 11 |
| | 10D9 | 36 | 0.048 | 1.3 |
| | 10E3 | 100 | 0.054 | 0.5 |
| | 11B8 | 170 | 0.17 | 1.0 |
| | 11C7 | 180 | 0.91 | 5.2 |
| | 12B6 | 110 | 0.00026 | 0.002 |
| | 12E10 | 83 | 0.23 | 2.8 |
| | 12F5 | 170 | 13 | 79 |
| | 15C4 | 210 | 0.13 | 0.6 |

positive signal is observed in spiked pools of normal human sera (NHS). In this study, a positive signal was defined as an intensity reading greater than 20 mm*mV using the Qiagen ESE Lateral Flow Reader. Due to sample variability, difference in assay performance was observed between each lot of pooled human serum. Six different pools of NHS were used to optimize each LFI prototype for nonspecific background signal. Then the pools were spiked with recombinant protein and used to determine the LOD of the LFI prototypes to account for possible matrix effects on signal intensity. The LOD of the 8F10/6F10 LFI was estimated to be 2 ng/mL when recombinant LcrV was spiked into NHS (Lot NHS207) (Fig 3A). The LOD of the 11C7/3F2 LFI was estimated to be 1 ng/mL when recombinant F1 was spiked into NHS (lot NHS207) (Fig 3B). Despite high analytical sensitivity to each antigen, differences of assay signal were

**Table 3. Assay sensitivity of the top 4 mAb pairs (LcrV or F1) by lateral flow immunoassay.**

| Antigen | Capture mAb | Detection mAb | OD (mm*mV) 1 ng/mL antigen | OD (mm*mV) Chase Buffer only |
|---------|-------------|---------------|----------------------------|------------------------------|
| LcrV | 8F10 | 6F10 | 56 | 0 |
| | 8F10 | 2B2 | 53 | 0 |
| | 8F10 | 6E5 | 51 | 0 |
| | 8F7 | 6F10 | 26 | 0 |
| F1 | 11C7 | 3F2 | 115 | 0 |
| | 11C7 | 15C4 | 93 | 0 |
| | 4E5 | 3F2 | 89 | 0 |
| | 10D9 | 3F2 | 57 | 0 |

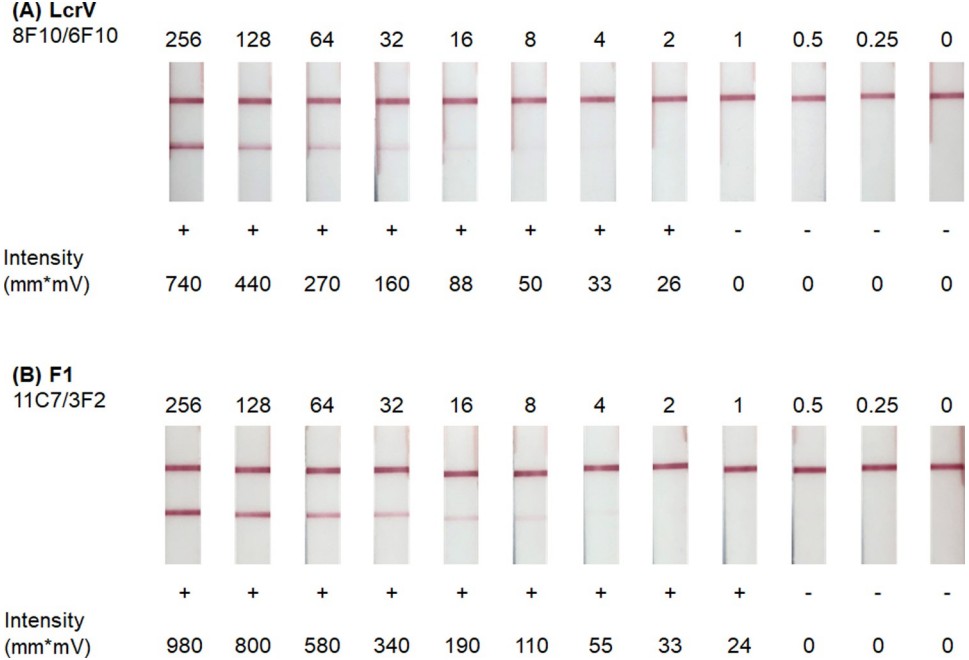

**Fig 3. Sensitivity of *Y. pestis* lateral flow immunoassays (LFI) using recombinant LcrV and F1.** LFI prototypes were tested with recombinant **(A)** LcrV and **(B)** F1 serial diluted into pooled normal human serum ranging from 0.25 to 256 ng/mL. Assay signal was evaluated and quantitated by optical density using a Qiagen ESE reader. Intensity $\geq 20$ mm*mV scores as positive.

observed between the six pools of NHS (S2 and S3 Figs). The aggregate LOD was determined as the lowest concentration which consistently resulted in positive LFIs among the pools of NHS. The LcrV and F1 prototypes produced LODs of roughly 1 ng/mL. Detection of both antigens in pooled NHS #28614 appeared slightly reduced (S2 Fig) and illustrates the variation in signal intensity that can occur when testing within different lots of pooled human serum.

## Generation of a multiplexed LcrV/F1 assay

To detect potentially all pathogenic isolates of *Y. pestis*, a multiplexed LFI was developed. The prototype was constructed with two test lines, one specific for LcrV (8F10/6F10) and one for F1 (11C7/3F2). Initial specificity testing of the dual LFI prototype was performed using bacterial lysates from Select Agent exempt strains of *Y. pestis*, clinically relevant near neighbors, and other Tier 1 bacterial Select Agents (Fig 4). Cell lysates were adjusted to an $OD_{600}$ of 0.5. *Y. pestis* Harbin-35 and KIM19 strains were positive for LcrV and F1. *Y. pestis* A12 Derivative 6, an LcrV⁻ strain, was positive only for F1. *Y. pseudotuberculosis* IP2666 lysate was positive for LcrV, but *Y. enterocolitica* was negative for LcrV. High levels of homology attribute to some degree of mAb cross-reactivity to LcrV of near neighbors [42,57]. Additionally, all other Tier 1 bacterial Select Agents were negative for both LcrV and F1.

## Antigen-capture ELISA development and optimization

The top eight mAbs ranked by LFI were used to develop a quantitative antigen-capture ELISAs for the detection of *Y. pestis* antigens. ELISA mAb pairs were initially evaluated using 1 μg/mL capture mAb and 0.1 μg/mL detection mAb. To compare the performance of each mAb pair, antigen concentrations at five-fold background signal were determined from a linear

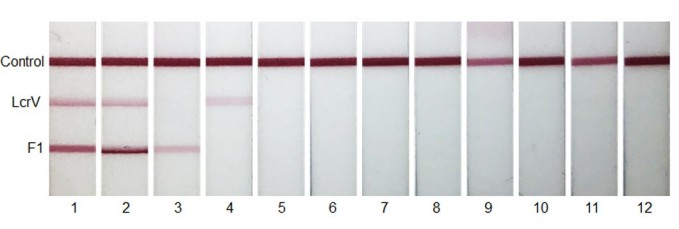

| # | Bacteria |
|---|---|
| 1 | *Yersinia pestis* Harbin-35 |
| 2 | *Yersinia pestis* KIM19 |
| 3 | *Yersinia pestis* A12 Derivative 6 |
| 4 | *Yersinia pseudotuberculosis* IP2666 |
| 5 | *Yersinia enterocolitica* WA |
| 6 | *Francisella tularensis* B38 |
| 7 | *Francisella tularensis* LVS |
| 8 | *Bacillus anthracis* Ames-35 |
| 9 | *Burkholderia pseudomallei* K96243 |
| 10 | *Burkholderia pseudomallei* 1026B |
| 11 | *Burkholderia pseudomallei* Bp82 |
| 12 | PBS control |

**Fig 4. Specificity testing of dual *Yersinia pestis* lateral flow immunoassay (LFI) against clinically relevant bacterial panel.** The dual LFI prototype containing test lines specific for LcrV (8F10/6F10) and F1 (11C7/3F2) was tested against a panel of bacterial lysates. Bacterial lysate (50 µL at $OD_{600}$ = 0.5) was applied onto the conjugate pad and chased with buffer. LFIs were imaged after 20 minutes.

regression plot generated from a two-fold serial dilution of each antigen (S4 Fig). This value (five-fold background) was selected to account for possible background signal caused by the mAb pairing. The top two performing mAb pairs for each antigen were optimized for capture and detection conditions by checkerboard ELISAs (Table 4). The theoretical ELISA LODs (defined as two-fold background signal) for each assay were determined using recombinant protein spiked into buffer and reported in Table 4 and S5 Fig. LcrV was detected at 74 pg/mL by 6E5/8F10 and F1 was detected at 61 pg/mL by 11B8/11C7.

## Discussion

The quick progression of plague infections warrants the need for sensitive, specific, and rapid diagnostic tools. Though most infections lead to bubonic plague, the more severe forms of the infection present with nonspecific clinical symptoms that can be challenging to distinguish from many other diseases. Commercially available LFIs have been developed for the detection of the F1 antigen and is used for patient screening, however no product is currently FDA approved to diagnose plague [18,20]. F1 encapsulates the bacteria and is shed; however, F1⁻ isolates have been identified and are fully virulent making the biomarker inadequate for diagnosing all plague infections [25,33]. To combat the potential of widespread infections, an RDT capable of detecting all pathogenic strains of *Y. pestis* is imperative. Through the isolation of a library of mAbs reactive to F1, as well as LcrV, we have developed assays for the detection of various pathogenic *Yersinia* species which may prove to be useful on multiple fronts.

The success of *Y. pestis* as a pathogen may be attributed to its genetic diversity. Widespread pandemics have been traced to three global biovars of *Y. pestis* (*antiqua, medievalis, and orientalis*). The *lcrV* gene is conserved among the biovars virulent in humans [54]. Polymorphisms in *lcrV* have been observed in the biovar *microtus* [58]. Some polymorphisms occur in critical

**Table 4. Limit of detection of enzyme-linked immunosorbent assays using recombinant LcrV and F1 antigens in buffer.**

| Antigen | Capture mAb | [Capture mAb] (µg/mL) | Detection mAb | [Detection mAb] (µg/mL) | LOD* (pg/mL) |
|---|---|---|---|---|---|
| LcrV | 6E5 | 2.5 | 8F10 | 0.13 | 74 ± 7.9 |
| | 6F10 | 2.5 | 8F10 | 0.063 | 75 ± 4.0 |
| F1 | 10D9 | 2.5 | 11C7 | 0.13 | 100 ± 41 |
| | 11B8 | 1.3 | 11C7 | 0.13 | 61 ± 2.0 |

*LOD defined as the concentration at two-fold background signal

regions impacting LcrV to form multimers, suggesting the attenuation of certain biovars in humans [54]. In addition to polymorphisms in *lcrV*, various isoforms of the F1 antigen exist due to point mutations [59,60]. The NT1 isoform (Ala$^{48}$ Phe$^{117}$) is most common and is expressed by the three global strains responsible for human infections [59]. Moreover, the adaptive immune response elicited by the NT1 isoform are cross-reactive to the NT2 (Ser$^{48}$ Phe$^{117}$) and NT3 (Ala$^{48}$ Val$^{117}$) isoforms suggesting these amino acid substitutions may not affect the binding of the described mAbs [61]. *Y. pestis* Harbin-35 and KIM D19 strains are included within biovar *medievalis* and maintain the three plasmids associated with virulence (pCD1, pPCP1, pMT1), but are attenuated due to the deletion of the *pgm* locus, which encodes a segment of the chromosome that includes a pathogenicity island [62]. The *pgm* locus encodes for various iron-regulated proteins necessary for bubonic and pneumonic plague infections [63]. Unlike most laboratory strains, Harbin-35 and KIM D19 maintain the pCD1 plasmid and LcrV expression. The A12 D6 strain is included within the biovar *orientalis* and is lacking the entire pCD1 plasmid. mAbs isolated in this study show reactivity to *Y. pestis* Harbin-35, KIM D19, and A12 D6 proteins which suggest mAb binding may be against all human pathogenic strains of *Y. pestis* (Figs 1–2 & 4). Further characterization of the mAb panel should be performed to confirm reactivity among all pathogenic *Y. pestis* strains.

Binding kinetics derived from SPR demonstrate that most of the mAbs produced in this study possess high affinity to their targets (Table 3). In general, higher affinity mAbs are preferred in immunoassays to achieve optimal analytical sensitivity which should result in improved clinical sensitivity; however, the mAb pairs which performed well in LFI and ELISA formats did not necessarily have the highest affinity or superior kinetics. Initial screening of LcrV mAbs by ELISA indicated that the pairing of high affinity mAbs 8F7 and 8F10 resulted in LODs 10 to 20-fold less sensitive than 8F10 and either 6E5 or 6F10, mAbs with lower degrees of affinity (Table 2; S4 Fig). Likewise, the mAbs used in the optimized F1 ELISAs (10D9, 11B8, and 11C7) were ranked among the middle for binding affinity ($K_D$). Additionally, mAb 12B6 had binding affinity over 2 logs greater ($K_D = 0.002$ nM) than the other mAbs in the panel but did not perform well in the antigen-capture format. Overall, the mAbs used to develop the sandwich assays all displayed dissociation constants less than 10 nM. These results exemplify that in addition to having high affinity, mAbs used in antigen-capture immunoassays require pairing synergy.

Interestingly, the top performing mAb pairs in the LFI format were not the same as the ELISA. Evaluation of LFI and ELISA mAb pairs for Ebola glycoprotein and *F. tularensis* in our laboratory have also shown differences between the two formats [64,65]. Though both LFIs and ELISAs are antigen-capture immunoassays, there are notable differences in the assay formats. ELISAs reach binding equilibrium with long incubation periods, minimize nonspecific binding with several wash steps, and are more sensitive due to enzymatic signal amplification. However, the ELISA format is less accessible in low-technology settings such as in the field or in developing countries due to laboratory equipment requirements and storage of temperature-sensitive reagents. In contrast, LFIs utilize capillary flow to apply reagents systematically without intermediate washing steps, are self-contained within a single dipstick, and produce results in less than 20 minutes by the colorimetric sensing of gold-nanoparticle aggregates, optimal for visual detection by the human eye.

The RDT immunoassay prototypes developed in this study had analytical sensitivities within appropriate levels for assaying serum. Levels of F1 in patient serum can range from 4–50,000 ng/mL [66]. Concentrations of soluble LcrV in human samples have yet to be determined, but a mouse model of infection indicates serum concentrations of LcrV are between 6–26.5 ng/mL 48 hours post-inhalational exposure [41]. To determine the diagnostic potential of LcrV and F1, further studies will need to be conducted within the 48 hours of exposure as

this window is crucial for administering treatment for patient survival The ELISAs developed in this study would be useful in quantifying LcrV and F1 in plague patient samples to further validate each diagnostic target. Additionally, the ELISA format may also be important in outbreak settings to allow for high throughput screening of patient samples. Furthermore, the top mAb pair for detecting LcrV by LFI was evaluated in a vertical flow immunoassay (VFI) format. The mini VFI prototype can detect as low as 25 pg/mL LcrV [67].

*Y. pestis* can invade the bloodstream and be detected within days of exposure, making blood a common test matrix among the three forms of plague. The LFI prototypes were optimized for serum as it is similar to whole blood and can cause matrix effects, or interference by components present in the clinical sample at the test line. Matrix effects were evident in testing both the LcrV and F1 LFIs as pools of normal human serum resulted in varying signal intensity. The LOD results for both the LcrV and F1 prototypes roughly 1 ng/mL. Additionally, the optimized LFI prototypes were able to analyze up to 50 μL of neat human sera with a minimal degree of pre-treatment. The tested assay protocol includes a sample preparation step in which mouse IgG was added to prevent HAMA interference. However, mouse IgG could be integrated into the sample pad prior to finalization of the LFI along with separation pads to allow for whole blood testing. Furthermore, the LFI prototypes described should be evaluated using other clinical matrices such as sputum or pus as these may be more clinically relevant for pneumonic and bubonic infections, respectively.

Plague remains a modern threat to public health, and LFIs are ideal tools for detecting and limiting the spread of infections. In addition to the risk of naturally occurring plague, *Y. pestis* has been used as a biological weapon. The use of *Y. pestis* in medieval siege warfare has been well documented, and in modern times, plague-infected fleas were disbursed by the Japanese army in China during World War II [68]. Several countries, including the United States of America and the former Soviet Union, have investigated the utility of *Y. pestis* as a bioweapon [9]. Further investigation of *Y. pestis* as a biological weapon is no longer permit under a treaty signed at the Biological Weapons Convention in 1972. Nonetheless, continued development of countermeasures against *Y. pestis* are warranted in defense of nefarious actions. Of concern is the intentional release of an F1⁻ strain as a bioterrorism tactic, as the F1 antigen is primarily regarded as a diagnostic and vaccine target [69–74]. The multiplexing of assays detecting LcrV and F1 increases the diagnostic ability of a *Y. pestis* RDT, as it may be capable of detecting all pathogenic *Y. pestis* strains including those found to be F1 deficient. The dual assay shows high specificity to *Y. pestis* without cross-reacting with other Tier 1 Select Agents with similar symptoms. It was anticipated that the LcrV/F1 prototype would be positive for LcrV *Y. pseudotuberculosis* as the homology is 97% compared to *Y. pestis* [75]. In cases where plague is suspected, a positive result for either F1 or LcrV should prompt immediate treatment [76]. More specificity testing needs to be performed using these prototypes with an increased panel of microbes known to present with similar symptoms. The prompt detection of all pathogenic *Y. pestis* is critical for minimizing casualties during naturally occurring outbreak or a potential bioterrorism act. Further evaluation on clinical samples collected from plague patients is needed to fully validate this multiplexed LFI.

While the immunoassay prototypes developed in this study were designed with patient samples in mind, a secondary use would be to evaluate animal populations and vectors for the presence of *Y. pestis*. As a zoonotic disease, plague surveillance requires the monitoring of natural animal reservoirs and associated vectors. Current methods of surveying wild populations include serological testing of sentinel animals, such as coyotes, which prey on smaller mammals [77]. Though coyotes do not present with symptoms, they do elicit an immune response to *Y. pestis* antigens [78]. The downside of serological surveillance is the persistence of antibodies long after exposure, meaning that a positive response may not be indicative of an active infection. Though expression of LcrV

and F1 are regulated by temperature, leaky expression of LcrV at 26˚C has been observed [79]. This leaky expression should be evaluated as a mean to detect *Y. pestis* in vectors such as fleas. Screening of small mammals and vectors could be conducted using an LFI in the field to gain more data regarding the prevalence of *Y. pestis* in these reservoir populations.

In addition to use in a diagnostic assay, exogenous antibodies have shown to be effective in protecting from and treating plague infections when administered pre- and post-exposure. In a mouse model of pneumonic plague, F1 specific IgG mAbs were able to confer protection when administered prophylactically [80]. Additionally, three human mAbs (one against F1, two against LcrV) isolated by naïve human phage displayed Fab libraries demonstrate some protection for bubonic plague in mice [81]. LcrV and F1 mAb cocktails do have synergy when administered together [82]. LcrV subunits form pentamers at the tip of the T3SS at the cell surface [36,83]. LcrV in this pentameric structure is associated with immunosuppression properties and have shown to elicit a more protective response in vaccine studies [37,84]. Testing the therapeutic potential of this large panel of *Y. pestis* mAbs in an animal model of infection may result in the development of additional treatment options for plague patients.

Monoclonal antibodies, specific to LcrV and F1, were used to develop antigen-capture LFIs and ELISAs with high analytical sensitivity. The LFI prototypes resulted in LODs of roughly 1 ng/mL for both LcrV and F1 when assaying antigen spiked into normal serum. The ELISAs were able to achieve an analytical sensitivity in the range of 61–74 pg/mL, at this point testing was only preformed in assay buffer. The detection of the F1 antigen is widely used to diagnose plague infections, however mutant strains of *Y. pestis* lacking the F1 antigen have been identified. Initial inclusivity/exclusivity studies of the multiplexed LFI demonstrates that the inclusion of a second antigen, LcrV, should improve the performance of the RDT. The LcrV antigen is crucial for virulence and may be used as an alternative marker of plague. Further evaluation of LcrV is warranted and can be accomplished using the tools developed in this study. In addition, side-by-side analyses of the assays developed here with other plague assays is a logical next step in order to rank performance. Finally, the immunoassays developed hopefully will be diagnostically useful and there could be potential of the mAbs being further developed into therapeutics, thereby assisting in the control of future plague outbreaks.

## Supporting information

**S1 Fig.** Top four LcrV LFI prototypes tested with **(A)** PBS or **(B)** *Y. pestis* Harbin-35 lysate for the detection of native antigen. Test lines were sprayed at 1 mg/mL and 5 uL of gold conjugated mAb ($OD_{540}$ = 10) was applied.
(PDF)

**S2 Fig. LFI prototypes (8F10-capture/6F10-detection) were tested with recombinant LcrV in six pools of normal human serum.** Each panel represent a different lot of pool serum from **(A-C)** Bioreclamation IVT or **(D-F)** Innovative Resources. Lot numbers are provided for each panel. Assay signal was evaluated and quantitated by optical density using a Qiagen ESE reader. Intensity $\geq$ 20 mm*mV scores as positive.
(PDF)

**S3 Fig. F1 prototypes (11C7-capture/3F2-detection) were tested with recombinant F1 in six pools of normal human serum.** Each panel represent a different lot of pool serum from **(A-C)** Bioreclamation IVT or **(D-F)** Innovative Resources. Lot numbers are provided for each panel. Assay signal was evaluated and quantitated by optical density using a Qiagen ESE reader. Intensity $\geq$ 20 mm*mV scores as positive.
(PDF)

**S4 Fig. Preliminary screen to identify top performing antigen-capture ELISA mAb pairs.**
Values shown are the concentrations of **(A)** recombinant LcrV and **(B)** recombinant F1 in ng/ml at five times background for each mAb pair. The values represent the mean of two independent ELISAs (each performed in biological triplicates).
(PDF)

**S5 Fig.** Antigen-capture ELISAs were performed to determine the limits of detection (LOD) for recombinant **(A & B)** LcrV and **(C & D)** F1. LODs were calculated using the linear regression of the optimized ELISA conditions to determine the concentration of recombinant protein in ng/ml at two-fold background. The values represent means of three independent ELISAs (each performed in biological triplicates).
(PDF)

**S1 Table. Primers for cloning LcrV and F1 genes from *Y. pestis* Harbin-35 into the pQe-30 Xa vector.**
(PDF)

**S2 Table. Summary of lateral flow immunoassay components evaluated.**
(PDF)

**S3 Table. Preliminary assay sensitivities of top mAb pairs by LFI for LcrV at 100 ng/mL.**
(PDF)

**S4 Table. Preliminary assay sensitivities of top mAb pairs by LFI for F1 at 100 ng/mL.**
(PDF)

**S5 Table. Preliminary assay sensitivities of top mAb pairs by LFI for F1 at 1 ng/mL.**
(PDF)

## Acknowledgments

We would like to acknowledge the guidance provided through the DCN Dx Custom Lateral Flow training for suggestions for optimizing LFI's.

## Author Contributions

**Conceptualization:** Derrick Hau, Brian Wade, Sujata G. Pandit, Marcellene A. Gates-Hollingsworth, Peter N. Thorkildson, Kathryn J. Pflughoeft, David P. AuCoin.

**Data curation:** Derrick Hau, Brian Wade, Chris Lovejoy, Sujata G. Pandit, Dana E. Reed, Haley L. DeMers, Heather R. Green, Emily E. Hannah, Megan E. McLarty, Cameron J. Creek, Chonnikarn Chokapirat, Jose Arias-Umana, Garett F. Cecchini, Teerapat Nualnoi.

**Formal analysis:** Derrick Hau, Brian Wade, Chris Lovejoy, Dana E. Reed.

**Funding acquisition:** Marcellene A. Gates-Hollingsworth, David P. AuCoin.

**Investigation:** Derrick Hau, Sujata G. Pandit, David P. AuCoin.

**Methodology:** Derrick Hau, Brian Wade, Chris Lovejoy, Sujata G. Pandit, Dana E. Reed, Teerapat Nualnoi, Marcellene A. Gates-Hollingsworth, Peter N. Thorkildson, Kathryn J. Pflughoeft, David P. AuCoin.

**Project administration:** Derrick Hau, Sujata G. Pandit, Dana E. Reed, Marcellene A. Gates-Hollingsworth, Peter N. Thorkildson, Kathryn J. Pflughoeft, David P. AuCoin.

**Resources:** Derrick Hau, Marcellene A. Gates-Hollingsworth, Peter N. Thorkildson, David P. AuCoin.

**Supervision:** Derrick Hau, Sujata G. Pandit, Dana E. Reed, Haley L. DeMers, Teerapat Nualnoi, Marcellene A. Gates-Hollingsworth, Peter N. Thorkildson, Kathryn J. Pflughoeft, David P. AuCoin.

**Validation:** Derrick Hau, Brian Wade, Chris Lovejoy, Sujata G. Pandit, Haley L. DeMers, Heather R. Green, Emily E. Hannah, Megan E. McLarty, Cameron J. Creek, Chonnikarn Chokapirat, Jose Arias-Umana, Garett F. Cecchini.

**Visualization:** Derrick Hau, Marcellene A. Gates-Hollingsworth, Peter N. Thorkildson, Kathryn J. Pflughoeft, David P. AuCoin.

**Writing – original draft:** Derrick Hau, Brian Wade, Marcellene A. Gates-Hollingsworth, Peter N. Thorkildson, Kathryn J. Pflughoeft, David P. AuCoin.

**Writing – review & editing:** Derrick Hau, Brian Wade, Chris Lovejoy, Sujata G. Pandit, Haley L. DeMers, Heather R. Green, Emily E. Hannah, Megan E. McLarty, Cameron J. Creek, Chonnikarn Chokapirat, Jose Arias-Umana, Garett F. Cecchini, Teerapat Nualnoi, Marcellene A. Gates-Hollingsworth, Peter N. Thorkildson, Kathryn J. Pflughoeft, David P. AuCoin.

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
