## [Decision Letter · Decision Letter 0]

8 Nov 2021

Dear Dr. AuCoin,

Thank you very much for submitting your manuscript "Development of a dual antigen lateral flow immunoassay for detecting Yersinia pestis" for consideration at PLOS Neglected Tropical Diseases. As with all papers reviewed by the journal, your manuscript was reviewed by members of the editorial board and by several independent reviewers. In light of the reviews (below this email), we would like to invite the resubmission of a significantly-revised version that takes into account the reviewers' comments. 

Editor comments:

LFI for detecting Yersinia pestis is often used in the field or at local health centres without any laboratory. Withdrawing and centrifuging blood are often not possible. Therefore you also should test other matrices such as sputum and pus collection.

Early detection of Yersinia pestis in serum is interresting. Could you give results on the monitoring of F1 and LcrV detection in samples of infected patients (or infected animals)?

We cannot make any decision about publication until we have seen the revised manuscript and your response to the reviewers' comments. Your revised manuscript is also likely to be sent to reviewers for further evaluation.

Sincerely,

Anne-Sophie Le Guern

Guest Editor

Javier Pizarro-Cerda

Deputy Editor

Editor comments:

LFI for detecting Yersinia pestis is often used in the field or at local health centres without any laboratory. Withdrawing and centrifuging blood are often not possible. Therefore you also should test other matrices such as sputum and pus collection.

Early detection of Yersinia pestis in serum is interresting. Could you give results on the monitoring of F1 and LcrV detection in samples of infected patients (or infected animals)?

Reviewer's Responses to Questions

**Key Review Criteria Required for Acceptance?**

**Methods**

-Are the objectives of the study clearly articulated with a clear testable hypothesis stated?

-Is the study design appropriate to address the stated objectives?

-Is the population clearly described and appropriate for the hypothesis being tested?

-Is the sample size sufficient to ensure adequate power to address the hypothesis being tested?

-Were correct statistical analysis used to support conclusions?

-Are there concerns about ethical or regulatory requirements being met?

Reviewer #1: Partly. see file attached

Reviewer #2: The objective of this study was to develop immunoassays for the detection of dual antigens (LcrV and F1) for the potential diagnoses of plague infections. The objective and study design was clearly articulated and the study design was appropriate to address the objective. No concerns over ethical or regulatory requirements were noted.

Reviewer #3: The objectives of the study is clear : to develop a dual Lateral Flow Immunochromatographic Assay (LFIA) for detecting Yersinia pestis in patients who suffer from plague infection by using 2 antigens F1 and LcrV. The novelty seems to be in the use of 2 antigens in a same test in order to limit the risk of false negative. Effectively the choice to add a second target LcrV is relevant as F1 is specific to Yersinia pestis but not expressed by all virulent strains. However LcrV is expressed by all virulent Yersinia pestis strains but homologs produced by others Yersinia species can hinder assay specificity. In this condition it would have been good to guide the final user in the interpretation of the test. What to do if both test line are positive, only F1, only LcrV or none of them ? in a context a patient seems to suffer from plague. What is the place of such test in the patient managment algorithm ?

The authors performed a huge development of mabs against FI and LcrV with the objective to develop Immunoassays LFIA and Enzyme-Linked ImmunoSorbent Assay (ELISA) and mainly an LFIA for detecting Yersinia pestis by targeting two different antigen F1 and LCRV. The characterisation of the selected hybridoma/monoclonal antibodies (mabs) was performed in the rule of the art with the determination of the subclass in order also to eliminate the IgG3 subclass, the realisation of western blot and the determination of the mabs affinity. It could have been useful to have information concerning the hybridoma culture as it seems that mabs are not produced via mousse ascite. Are they produced in vitro in serum free medium or not ? what about the condition of culture ? and what about the yield of mabs production (mg/ml) before and after purification ? This information is key for the future of the test and to be able to evaluate the feasibility to produce it at large scale. The information that selected hybridoma are good or bad producers of mabs is key to switch from research and developement to industrilization and production. All of this is important to be sure that such mabs could impact positively the patient care. One way to accelerate their transfer to production is also the adaptation of the hybridoma to serum free medium as it is more or less mandatory for industrials partners due to the fact selection of feotal calf serum or donor calf serum induce batch selection and is also costlty and time consuming.

The authors also performed an important screening of the best pairs of mabs for LFIA and ELISA development known to be fastidious and intensive work. However it would have been necessary to have a better understanding of the intended use of the test. It is not clearly described and this is reflected by the implementation of the development and format of test developed by the authors. Moreover, it would have been a benefit for the development of the dual antigen Yersinia test to describe a comprehensive target product profile (TPP). The TPP needs to provide details on the minimum and optimal performance and operational characteristics of the diagnostic tests to be developed. Researchers, developers, and manufacturers use TPPs to ensure that R&D activities are focused on relevant products and designed for the contexts and needs of end-users. There is no TPP described on this study. This TPP include among others the choice of the patient sample nature on which the test need to be developped. In this case the authors proposed the serum but this choice is not so much argued except the fact Yersinia pestis can quickly invade the bloodstream and be detected within days of exposure and consequently make serum an ideal matrix to assay bubonic, pneumonic and septicemic plague. In this case why do they not target whole blood ? moreover why not target sputum or saliva ? Does sera is not too late as we understood that treatment need to be delivered within around 24H00 after exposure. What about the possibility to use plasma (in this case the impact of the anticoagulant need to be explored). Serum seems to be too much restrictive except if the authors can justify this choice apart from the ease of access to develop. Serum need to be used during the devopment of the test but you need to evolve to whole blood. We do not feel that is was an objective for the authors. Serum need to be explored but in this study it seems that the developement of the test is too much deconnected from whole blood. Finally the final formulation must work with whole blood, serum and plasma.

There are also in this study no comparison with commercialized tests that would have been mandatory as they tried to demonstrate an improvment via their test vs the existing ones.

It seems that the authors improve the LoD in analytical condition on recombinant antigen but to definitly proof it would have been better to perform an experimental comparaison instead of a bibliographic one and not only by using recombinant but also on strains and sample from patient. In addition, if you do not carry out an analytical comparison on the same recombinant antigens and on a same format of test and same implementation of it, it is difficult to conclude on the performance improvement. Finally, the most important evaluation is that on patient samples because the improvement on recombinant antigen is not necessarily predictive of the improvement on patient samples.

It lacks also the development of an algorithm to vizualise how this test could be used vs the gold standard and others technics whatever you are in the field or in a labs. As an example is it a screening test that need to be confirmed or to use it purely for diagnosis ? you will not develop the test in the same way and the raw material adjustement will not be the same as you need to know if sensitivity is prefer over specificity or if both need to be at their maximum/ideal value near 100%. We do not kow what is acceptable and we have no information on the minimal performance. This is important as it allows during the development phases to adjust the quantity of each raw materials in order to get the best balance between sensitivity and specificity depending you or more on a screening or diagnosis mode. All of this is a prerequisite to develop a test and not sufficiently highlighted. It is also important depending of the prevalance to have an objective concerning the positive predictive value and/or negative predictive value to be achieved that is fonction of sensitivity, specificity and the geographical prevalence. The authors did not develop also such important topic.

The choice to add a second target is relevant as F1 is specific to Yersinia pestis but not expressed by all virulent strains. However LcrV is expressed by all virulent Y.Pestis strains but homologs produced by other yersinia species can hinder assay specificity. This oblige to guide the final user in the interpretation of the test and what they need to do in case for example you have only positivity for LcrV that could be induced by non specific species. A quick guide could have been suggested by the authors.

Again, the authors developed on serum instead of on whole blood independently of the fact that ideally the final test could be used not only on whole blood or capillary blood from fingers but also on plasma and sera. We understood through the article that the test need to be used on the field near patient but as we have no TPP and no algorithm of it implementation it is important that the authors justify such choice. If whole blood is targeted the component of the strip and/or composition need to be adapted to such matrix in order not to allow red blood cell to go through the membrane by using mabs against red blood cells or a porosity that keep them in the sample pad. The serum can be used but in this case its volume and way of implementation need to be think with a final use with whole blood that seems not to be the case as they used a chase buffer, dipstick format and so on. The choice of the authors to use a dipstick format and the use of a chase buffer is not aligned with the final use of such antigen test. The use of both is usually of what is expected or more adapted for IgG/IgM detection as we usually used small volume of whole blood (around 10µl) or serum (around 5µL) that could fit with dipstick format and the necessity in this case to use a chase buffer as you absolutly need it to allow the migration of the patient sample. The choice to use low volume sample in this condition is one way to avoid hook or prozone effect. For antigen detection you can use more volume in order also to improve the sensitivity with the limit of drops of whole blood you can get from the tip of the fingers. But again if you are able to get sample by performing venipuncture volume is not limitant and the less number of step you have the better it is. Consequently chase buffer is usuallly not use for antigen detection. It is the reason why TPP is important as such information are included into it as well as information concerning the final implementation of the test. The casette format is also more appropriate to the field as you do not need to use a tube and above, all nobody can influence the migration of the sample into the strip during the time of migration as usually people takes the strips for observation and take out and in the strip inside the tube during its migration. To come back to the use of a chase buffer and its impact. For antigen detection we usually do not use a chase buffer, in this experimentation 150µL of chase buffer are used, as the volume of serum is not sufficient to allow its migration from the sample pad to the adsorbent pad. The volume of serum used in this case 40 µl plus 150 µl of chase buffer will impact also the level of the interference that could be reduced by using this chase buffer in a context also where the authors used pool of serum that mask potential high interference from one serum of the pool. The choice to use a chase buffer is not without consequence on the choice of the components of the test and in particular that of the nitrocellulose membrane and on the entire formulation of the test.

It would have been better that the authors developed the test by using a casette format. They can start the developement in dispstick but with the appropriate volume of sera and no chase buffer but in all cases a cassette need to be used to continue the development in order to be as soon as possible near the final format of test that will be used on the field. The integration of a strip in a cassette is not without consequences on the choice of components, their thicknesses… as there are key contact at the interface of the strip and the cassette which will condition the quality and speed of migration of the sample, of the conjugate in the whole test and consequently the sensitivity and specificity.

The determination of analytical performance could have benefited of the reader used to read the strips. This opportunity was not seized and could have shed important light on the variability of the tests. The creation of a pool in this case could have been more relevant for multiplying the number of strips per point of concentration in order to determine a Limit of Blank (LoB), a limit of detection (LoD) as conventionally implemented for the determination of the performance of a quantitative method. We understand here that this is a development of a qualitative test but the use of the reader would have been decisive to determine an LoD rather than to estimate its variability through different pools which from a biological point of view have no reality included for non specific binding or cross reactivity study. It would have been a criteria of differentiation compared to others. To provide a standard method for determining LoB, LoD and LoQ, Clinical and Laboratory Standards Institute (CLSI) has published the guideline EP17 and protocols for determination of LoD and LoQ. LoB,LoD and LoQ are terms used to describe the smallest concentration that can be reliably measured by an analytical procedure. The LoB is the highest apparent analyte concentration expected to be found when replicates of a blank sample containing no analyte are tested. LoB = meanblank + 1.645(SDblank). To establish it, the sample type to use is a sample containing no analyte, e.g. zero level calibrator and to perform 60 replicates and 20 for verification. The sample characteristics are negative or very low concentration sample that is commutable with patient specimens. The LoD is the lowest analyte concentration likely to be reliably distinguished from the LoB and at which detection is feasible. LoD is determined by utilising both the measured LoB and test replicates of a sample known to contain a low concentration of analyte. LoD = LoB + 1.645(SD low concentration sample). The samples containing low concentration of analyte, e.g. dilutions of lowest concentration calibrator again 60 replicates to establish and 20 to verify. The samples charcateristics are low concentration samples, such as a dilution of the lowest non-negative assay kit calibrator or patient specimen matrix containing a weighed out amount of analyte, commutable with patient specimens. LoQ is the lowest concentration at which the analyte can not only be reliably detected but at which some predefined goals for bias and imprecision are met. The LoQ may be equivalent to the LoD or it could be at a much higher concentration but not so much relevant to dertermine in this cwork as the final test will be a qualitative one. For a statistical point of view this would have been of great interest whatever it is for the LoD determination of the ELISA or LFIA.

The use of serum plus a chase buffer and the fact that the authors made pool of serum do not allow to well evaluate the risk to get false positive. The pools of serum have no clinical reality in particular to study non specific binding or interference due to human anti mousse antibody or others factors that induce it. It would have been more relevant to go through more individuals sera to measure the risk of false positive due to particular composition with the presence of human anti-mouse antibody, rhumatoid factor, anti-nuclear antibodies or auto antibodies. Moreover we have no information concerning the serum used and the fact they could have been heat inactivated or pre-treated, conditions that induce usually less interferences. To study that, it is better to use specific pannel known to interfere with test developement and test them on the best pairs of antibodies that give the expected sensitivity and to go more through individual sera to estimate the risk of false positive before to add any treatment of the test. It would have been also important to supply sera on others pathologies as people that could suffer from Yersinia pestis have been multiinfected. Sera from countries in which the test will be deployed is better as sera from such patient are more complex that those from developed countries and interferences could be higher.

Information is lacking on the effort made by the authors to set up the conjugate (determination of pH, load of mabs at the surface of the gold nanoparticles) Did they used the salt method ? but also to set up the nitrocellulose membrane (pH and load of mabs per test line), quantity of conjugated per strip… It is difficult to assess if these results could be improved not only for a sensitivity point of view but also to reduce the risk of false positive due to interference. In development we usually test multiple optical density of conjugate vs different quantity of mabs on nitrocellulose in order to determine the optimum for a sensitivity and specificity point of view. Finally do the authors work at fixed concentration or do they perform adjustement, if not it is one way to improve the test.

It seems that the authors did not try to test more complex combinaison by using 3 or 4 mabs that could sometimes improve the performance of the test. It’s something that need to be investigated even if however it is better for a product managment better to have a minimum of different mabs.

The tests are read at 20 minutes. How this time to result has been determined? is this the optimum ? A kinetics could have been useful to ensure that the signal on the test line does not change over time in this case after 20 minutes. It could have been good to have the reading result by eye with one or two operators in order to establish the LoD as the final test will not be read by the reader. This could also be more discussed.

The positioning of the test lines has not been discussed first F1 and LcrV in second position before the control line, is it an optimum ? what about results obtained by a reverse positioning ? is there any rational to put first F1 and LcrV in second position ? The positionning of the test line could also influence the LoD of each target.

The ELISA could be also very usefull first to perform the verfification performance of the LFIA test and also in a laboratory environnment in addition to the PCR or alone for laboratory not equiped to perform PCR for many reason. The ELISA has not been evalauted in serum but in PBS, how the authors justify this choice ? it would have been better to compare the LoD of the LFIA vs ELISA on a same standard range with F1 an LcrV spiked in a same matrix in this case serum. Moreover did the authors tried to mix F1 and LcrV in a same sample in order to determine if it impacts the LoD of each in this condition. 

The bacterial strains were used from the point of view of detection and specificity and not from the point of view of LoD which would have strengthened the analytical study performed by spyking the serum with recombinant F1 and LcrV. It would have been usefull to determine the number of bacteria that the test can detect in a context we have no more data in this study based on samples patient analysis. It seems also that LcrV level in patients is not so well defined …so determination of the smallest quantity of bacteria detectable by the test is important.

Overall, the authors made a huge work to generate mabs against F1 and LcrV in order to detect Yersinia pestis strains by LFIA or ELISA. The developemnt plan is not enough point of care oriented as whole blood was not enough in mind whatever it is for the component used for the strip, or the format like dipstick format. The problem is not that the authors performed part of the development by using sera but more that they used a chase buffer. Part of the analytical performance could be done by using sera but not in the conditions determined by the authors. The determination of the analytical sensitivity need to be performed again in an appropriate format of test and with more strips for each concentration of the standard range in serum and in whole blood by spiking F1 and LcrV but also need to be established by using a standrad range of different concentration of bacteria. The concentration of ng/ml of F1 or LcrV is required but not sufficient they need also to perform a standard range with different bacteria species and to determine the lower quantity of bacteria the test is able to detect. However In this study we have no true study of non specific binding, crossreactivity as the use of pool do not allow to study such topic. For a statical point of view the authors did not use the opportunity of the reader. We have also no comparison with commercialized test. Finnaly the way in which the mabs and the formulation of the prototypes could be valued was not addressed by the authors in the discussion and this is an important point in determining whether all this work can benefit to the patients quickly. The conditions of access to the mabs and the conditions of availability of the tests were not discussed. Accessibility is key for users and patients from developing countries because too often tests are too much expensive.

**Results**

-Does the analysis presented match the analysis plan?

-Are the results clearly and completely presented?

-Are the figures (Tables, Images) of sufficient quality for clarity?

Reviewer #1: partly. see file attached

Reviewer #2: The results are clearly and completely presented. The tables and figures are of sufficient quality for clarity.

Reviewer #3: The results correspond well to the plan presented and the quality of the tables, photos and figures are relatively of good quality. However some points need to be discussed.

Table. 1, the objective of the double F1-V immunization has not been sufficiently described in the context of human diagnostic use, nor does the low number of hybridomas selected under these conditions. The F1-V immunization seems to have brought little mab with the exception of 2B2 recognizing LcrV and 3A2 recognizing F1 but which were not kept in the formulation described as being the best performing pairs. Moreover, 2B2 and 3A2 are specifics for LcrV and F1 respectively, but does the authors perform experimental study to verify it.

We have no particular remark regarding the nature of the experiments carried out to generate or characterize the mabs. However, Fig. 1A/B and Fig. 2A/2B, the authors could have discussed the fact that for the detection of the F1 protein the intensity of the bands in reducing condition is among the weakest for the antibodies selected in the best Pair 3F2 and 11 C7 which tends, in contrast to 3A2, to a lower recognition of the monomeric forms. Does The authors have an explanation for this and what could be the impact ? This is not the case for the best pairs determined with regard to the capture of LcrV for the 2B2 and the 6F10 with the exception may be for the 8F10 whose signal intensity seems to be weaker than that of the 2B2 and 6F10 in reducing condition or not. Moreover, it would have been relevant to carry more western blots with others strains than Yersinia Pestis harbin-35.

Table. 2, with regard to the dissociation constants, it is often observed that the antibodies of better affinities are not necessarily the antibodies that are found in the best combinations of antibodies in capture and in revelation as could be observed by the authors. It is in all cases important to benefit from this characterization upstream of a first screening while remaining careful not to rule out too quickly antibodies whose affinity could be assessed as low on the basis of this characterization

Page 17 we understand that the best performing pair (capture/detection) against native LcrV was 8F10/2B2 but that due to consistent nonspecific reactivity the authors switched on 8F10 / 6F10. They refers to supplemental figure 1 where the result in the photo are almost negative for the couple 8F10/6F10 on the lysate of Y pestis Harbin-35 with better results for 8F10 / 2B2 couple. Contrasting also with results in Table 3 that show equal sensitivity for 8F10/6 F10 compared to 8F10/B2. Does the authors could explain this ? We can not measure also the effort done by the authors try to reduce interferences with 8F10/2B2 pairs that seems to be more sensitive vs 8F10/6F10. In this context it would be important to know if the authors made any adjustments of these conjugates vs the antibody load on the nitrocellulose membrane? This work could have allowed to probably keep more combination to be tested on the samples of patients in a context where this combination 8F10 / 2B2 does not seem to interfere with PBS and where the use of a serum pool to assess non-specificity is irrelevant because in practice it is carried out on a large number of individual serum, all coming from or known to interfere.

Fig.3 the quality of the photos is good but the reading with people who are not rapid test developer would have been appreciated vs the measurement with the reader in order to have a better idea of the LoD that can be determined by a neophyte in the field. Developers will see as well as a reader, which may not be the case with uneducated people and more representative of users in the field.

With regard to Fig.4, it would have been preferable to present the expected results for each of the strains vs. the results obtained. How does The authors explain the negative result for LcrV for the strain Y pestis A12 derivative 6 described to be a strain expressing LcrV ? it does not appear that the discussion provides an explicit answer to this question. Moreover, is the negative result for F1 for strain IP2666 is expected ? if not what is the explanation? If the expected positive results are negative is it linked to a lack of sensitivity at this concentration ? what about the result with a higher quantity of bacteria? For the enterocolitica strain, it is specified that the result is negative for LcrV, nor does it seem much more positive for F1. Can the authors provide more details? It seems that the results obtained for the other strains are as expected, ie negative.

ELISA development could perhaps have been benefited from antibody screening less dependent on that carried out in LFIA, in particular for the detection of the F1 antigen. The 6E5/8F10 combination allowing the detection of LcrV is one of the combinations identified in LFIA, without however, that we can clearly understand why it was not retained in LFIA nor indeed the efforts made to maintain it, as others, besides vs 8F10 / 2B2 or 6F10. The authors mentioned page 19 an LOD of 61 pg/ml obtained by an 11B8/3F2 pair in ELISA and refers to table. 4 in which this combination is not mentioned because only 11B8/11C7 appear. Could The authors clarify this? as in supplementary fig 4 11B8/3F2 or the reverse does not seem to be in the best combination. If this is an error ? that allows us to identify that this combination has not been tested in LFIA and could be interesting to test. Even though the combinations are different, they have at least one antibody in common as the 8F10 is in capture in LFIA whereas it is in revelation in ELISA idem with regard to the detection of F1 for the 11C7. In any case, it is not surprising that the combinations of antibodies were different, sometimes they are even further apart or have a different number of antibodies.

Why the authors did not determine the LoD of the ELISA in serum just like for the LFIA ? This induces a non-negligible bias and rather in favor of the ELISA. Also why the strains have not been tested in ELISA. Here again, a determination of the smallest quantity of bacteria detected in ELISA and LFIA would have been a good complement to this study by being closer to patient conditions. The development of ELISA is as well as important as it could have a positive impact on the field. It allows as mentionned in the discussion an hightroughput implementation in laboratory in order to diagnose a lot of patient by using automation fluidhandling device. It could be of interest during a pandemia.

**Conclusions**

-Are the conclusions supported by the data presented?

-Are the limitations of analysis clearly described?

-Do the authors discuss how these data can be helpful to advance our understanding of the topic under study?

-Is public health relevance addressed?

Reviewer #1: partly. see file attached

Reviewer #2: The conclusions are supported by the data presented. The public health relevance was adequately addressed.

Reviewer #3: The conclusions are supported by the data presented but all the limitations of analysis are not clearly described as some of them are due to the methodology excepted the absence of data on sample from patients clearly mentionned by the authors. They mentionned also that ELISA was only tested in spiked PBS. It seems that they improve the LoD compare to commercially available test but we have no comparaison in this study with commercialized one. it could have been interesting to have a comparison of the prototypes vs the tests marketed.

The authors clearly describe the benefit of such test in different context like surveillance in order to prevent outbreaks, to have tools to detect quickly Y.pestis in a context of bioweapon, to evaluate animal populations and vectors for the presence of Y.pestis and the fact they envisage also a potential use of such mabs for protecting from plague infection and for a therapeutic purpose with however no information of their capacity to neutralize the bacteria in this article. We estimate that at this stage the prototypes developed need more development including in analytic condition and before to be tested on a huge volume of samples from patients. Sure that the mabs seems to have a huge potential for such development. It will be a pity not to continue such developement. For a public health point of view the authors need to develop more how they envisage to continue the development and establish the clinical performance of such prototypes. The less sample from patients they will test during the development the more prototypes with different combinaisons of mabs they will need to test to determine the clinical performance. Selection of mabs pairs only by using recombinant or strains is not suficient to predict the performance on sample from patients. It is the reason why it is better to develop the test near the patient whatever the country. Beyond this we need to understand how they envisage a valorization with a product accessible to countries in development with a uniform performance from batch to batch. The fact they wish to publish is good to inform the feasibility to get such dual antigen test but it is not sufficient as the most interresting is how the authors will make it possible on the field to better manage plague outbreak. The establismhent of such plan is of interest.

The limitation of this study is not related only to the lack of results on sample from patients but also to the method used to develop these first prototypes. Furthermore, we can not visualize the effort made by the team to adjust the quantity of mabs in detection or in revelation, which would certainly make it possible to have more possible combinations to test on patient samples. Furthermore, a sensitivity and specificity study cannot be carried out without an adjustment phase. This adjustment phase balances the test to achieve the expected results on these two criteria. It seems that the authors have worked with equal amounts of conjugate and antibody on the membrane which will de facto rule out possible combinations. It seems that they made a choice of combinaison with no study on interferences appearance with individuals serum but more on sensitivity. The two are too much deconnected and the choice of the best combinaisons need to be performed not only on sensitivity criteria but also on specificity. The overall performance of the test is finnaly also influenced by the type, volume of sample and the type of implementation with chase buffer or not in a dispstick or casette format and as the team. The all forms an almost inseparable set of parts. As soon as you modify a parameter the whole performance must be re-evaluated. We do not perceive in this development the parameters which influence more the performance of test. We need to better understand in what context this test will be used. In addition, the development of the LFIA test must be reviewed taking into account that the matrix chosen to allow its use as close as possible to the patient need to be a whole blood matrix and that the test format need to be a cassette format for a better convenience and safety both for the operator and to guarantee the correct performance of the test. This does not prevent that the final test must be able to be used not only in whole blood but also in serum and plasma and that the desired performance of the analytical and clinical test must be achieved in each of these matrices. This means that in the context of this study, the performance established in serum by the team must therefore again be the subject of a study under conditions of implementation of the test more appropriate to the field. This new implementation may impact the work of selecting mabs to determine the best pairs which in any case have not been sufficiently explored from a repeatability point of view for both sensitivity and specificity. In all cases, the sample volumes must be compatible for each of the matrices with an implementation of the test without a chase buffer. Part of the evaluation could be done in serum and then confirm via spiking in whole blood but with volumes allowing these matrices to migrate without chase buffer with as a prerequisite for whole blood to block the red bood cells in the sample pad by playing on the porosity of this component or with the addition of mabs against red blood cells. This LFIA test should be further evaluated with individual sera to better determine the risk of interference, but also in whole blood with adifferent anticoagulants for plasma investigation.

It is globally difficult to evaluate the ways of optimizing this test because we do not have access to all the experiments, in particular if there has been a great effort to carry out on the adjustments of the conjugates and the quantities of mabs on the nitrocellulose membrane. We do not have more information if a sample volume study was generated and if it strongly impacted the sensitivity/specificity balance. Depending on the end-use context of the test, the sample volume can be adapted and participate in improving the performance of the test, hence the importance of testing several sample volumes both for the sensitivity study and specificity. One of the limitations of the study recognized by the authors is the absence of results on well characterized samples from patients and it is all the most important to verify that the feasibility of assaying LcrV in patients remains a question because the concentration levels during infection do not seem to be fully understood in humans. This is not the case for the dosage of F1 in patients where the concentrations are described between 4 and 40,000 ng / ml. A study in mice shows concentration levels of 6 to 26 ng / ml. If it turns out that the concentrations in humans of LcrV are of this order of magnitude, the development of the ELISA in serum and plasma is important to characterize the samples and as back up in case the LFIA test lacks of sensitivity. From a public health point of view, it is important that the authors mention how this test will be accessible for those who need it.

A nice article could be generated with the determination of an analytical and clinical performance carried out according to the rules of the art in ELISA and LFIA with in addition for each of these formats the development of an IgM detection test and IgG since the teams have the recombinant antigens. The development of a combo LFIA dual antigen test plus a strip for an IgG/IgM detection could also be a good way to catch cases not detected via the antigen test but via IgM level and in any case very useful for carrying out epidemiological studies in the field in order to best map the epidemics.

Beyond the development of the LFIA test, the development of the ELISA is not sufficiently highlighted, probably again due to lack of TPP. We have no analytical performance in serum either via the recombinants or with the bacterial strains either from a sensitivity or specificity point of view. The TPP need to be described to guide the development of the LFIA test and the ELISA test which could probably be more used in confirmation when PCR is not available or when breaks in raw materials happen as it was the case during the sars cov 2 pandemia.

**Editorial and Data Presentation Modifications?**

Reviewer #1: (No Response)

Reviewer #2: Just some minor editorial suggestions:

Line 38: Full stop omitted after the Y for Y. pestis

Line 106: Define F1 minus strain at first use. 

Line 117: Expression of F1 is temperature-induced at 33C and above.

Line 171: The word female should not be a capital. “Twenty Female CD1 mice..”. 

Line 179: The word “or” should be “for”. “to account or variability”

Line 222: BSL2 should be BSL3. “biosafety level 3 (BSL2)”

Line 224: “Francisella tularensis” should be abbreviated to “F. tularensis”. First use of full scientific name was in line 218.

Line 225: “Bacillus anthracis” should be abbreviated to “B. anthracis”. First use of full scientific name was in line 219.

Line 246: Write out SDS. 

Line 262: “ten” should be in numerals. Numbers from ten and above should be written as numerals.

Line 266: “40uL”. Don’t start a sentence with numerals.

Line 294: “2” should be written in words. Numbers from zero to nine should be written in words.

Line 349: “3” should be written in words. As above.

Line 603-809: Check reference list. Full names of authors not listed for a number of publications i.e. references 17 (line 639), 20 (line 648), 21 (line 651), 33 (line 685), 47 (line 725), 54 (line 745), 58 (line 756), 59 (line 760), 62 (line 768), 65 (line 775), 66 (line 778), etc. 

Lines 205, 227, 238. 259, 260, 262, 263, 399, 432: There should be a space between the number and the unit of measure e.g. line 205 - 450 nm, line 259 - 1 mg/mL etc.

Reviewer #3: (No Response)

**Summary and General Comments**

Reviewer #1: (No Response)

Reviewer #2: The manuscript describes the development of a dual target (F1 and LcrV) lateral flow immunoassay for detecting Yersinia pestis. There are publications describing the development of single target (F1 antigen) RDTs, but not a dual target LFI. To my knowledge, there is only one commercial RDT targeting dual antigens (F1 and V), but it is not approved for human use. 

The LFI developed in this study was only tested against simulated serum samples. As mentioned by the authors, further evaluation of the multiplexed LFI is needed on clinical samples collected from plague patients. The paper is well written and adds to the body of knowledge on the topic.

Reviewer #3: The development of the LFIA and ELISA tests is not far enough and results obtained too much preliminary. The methodology used to develop the tests whatever it is the LFIA or ELISA should be reviewed and should start with writing a TPP in order to define the ideal test needed on the field and/or in the laboratory (cf previous sections for more details).

This work is not completed enough to allow a third party to perform a proper valuation of these prototypes on samples from patients. The authors need to continue their effort in order to accelerate the provision of a sufficiently developed prototype, adapted to the field with a well defined analytical performance (sensitivity and specificity) on recombinants and strains in order to use the precious and rare samples from patients on the best prototypes to verify its clinical performance. Moreover the authors will have to ensure very quickly the performance of the prototypes under development on samples from patients and it seems crucial for the detection of LcrV. It is even preferable to develop on samples from patient but probably the constraints to develop test for plague on such samples are high. The valuation of this work will go through the provision of mabs and the formulation of the prototypes developed and this point is not sufficiently addressed in the discussion. The team need also to describe the collaboration they need to build in order to establish the clinical performance of their prototypes and its added value vs commercialized available tests. Finally in case they are successful they need to establish an access plan in order that patients can benefit of such test that can save their life by being diagnose earlier and therefore treat earlier.

PLOS authors have the option to publish the peer review history of their article (what does this mean?). If published, this will include your full peer review and any attached files.

Reviewer #1: No

Reviewer #2: No

Reviewer #3: No
---

## [Decision Letter · Decision Letter 1]

28 Feb 2022

Dear Dr. AuCoin,

We are pleased to inform you that your manuscript 'Development of a dual antigen lateral flow immunoassay for detecting Yersinia pestis' has been provisionally accepted for publication in PLOS Neglected Tropical Diseases.

Best regards,

Anne-Sophie Le Guern

Guest Editor

Javier Pizarro-Cerda

Deputy Editor

Reviewer's Responses to Questions

**Key Review Criteria Required for Acceptance?**

**Methods**

-Are the objectives of the study clearly articulated with a clear testable hypothesis stated?

-Is the study design appropriate to address the stated objectives?

-Is the population clearly described and appropriate for the hypothesis being tested?

-Is the sample size sufficient to ensure adequate power to address the hypothesis being tested?

-Were correct statistical analysis used to support conclusions?

-Are there concerns about ethical or regulatory requirements being met?

Reviewer #1: -Are the objectives of the study clearly articulated with a clear testable hypothesis stated?

-Is the study design appropriate to address the stated objectives?

more or less.

The authors have not addressed the main criticism : they did not use the antigens in their native state for the evaluation of their dual LFIA. They have used bacterial lysates thermally inactivated (80°C, 2h) or recombinant proteins. Sensitivity of the tests is thus based on proteins that do not reflect the native proteins. Evaluation of the performance of a test (and particularly the sensitivity) is a major point when this type of tests is developed, and must be as close as possible of the natural native antigen.

**Results**

-Does the analysis presented match the analysis plan?

-Are the results clearly and completely presented?

-Are the figures (Tables, Images) of sufficient quality for clarity?

Reviewer #1: see above

**Conclusions**

-Are the conclusions supported by the data presented?

-Are the limitations of analysis clearly described?

-Do the authors discuss how these data can be helpful to advance our understanding of the topic under study?

-Is public health relevance addressed?

Reviewer #1: see above

The

**Editorial and Data Presentation Modifications?**

Reviewer #1: (No Response)

**Summary and General Comments**

Reviewer #1: The main criticism was not addressed by the authors. The article is based on the performance evaluation of a test that could be used for human clinical field diagnosis. The sensitivity of the test is evaluated with antigens that do not reflect the reality of antigens that would be present in real life, nor are there any clinical samples.

The authors argued that they could not use culture supernatants with the excuse that the culture supernatants do not contain enough antigen. I know (because I have already done it) that the bacteria Y. pestis after growing can be centrifuged, resuspended in a small volume of medium or PBS to concentrate them. After pipetting of the bacterial pellet back and forth several times, the F1 and LcrV antigens (that are not tightly attached to the bacteria) are found soluble in the supernatant and can be harvested after centrifugation (they are concentrated in the supernatant (which can then be filtered for safety).

PLOS authors have the option to publish the peer review history of their article (what does this mean?). If published, this will include your full peer review and any attached files.

Reviewer #1: No

---

## [Editor Report · Acceptance letter]

18 Mar 2022

Dear Dr. AuCoin,

We are delighted to inform you that your manuscript, "Development of a dual antigen lateral flow immunoassay for detecting Yersinia pestis," has been formally accepted for publication in PLOS Neglected Tropical Diseases.

Best regards,

Shaden Kamhawi

co-Editor-in-Chief

Paul Brindley

co-Editor-in-Chief
